# Recent Advances in Designing Adeno-Associated Virus-Based Vaccines Against Viral Infections

**DOI:** 10.3390/pharmaceutics16111360

**Published:** 2024-10-24

**Authors:** Njabulo Mnyandu, Ridhwaanah Jacobs, Patrick Arbuthnot, Mohube Betty Maepa

**Affiliations:** Wits/SAMRC Antiviral Gene Therapy Research Unit, Infectious Diseases and Oncology Research Institute (IDORI), Faculty of Health Sciences, University of the Witwatersrand, Parktown 2193, South Africa; njabulo.mnyandu@wits.ac.za (N.M.); ridhwaanah.jacobs@gmail.com (R.J.); patrick.arbuthnot@wits.ac.za (P.A.)

**Keywords:** adeno-associated viral vectors, vaccines, viral infections, SARS-CoV-2

## Abstract

Over 80% of the world’s deadliest pandemics are caused by viral infections, and vaccination remains the most effective way to prevent these infections from spreading. Since the discovery of the first vaccine over two centuries ago, several vaccine design technologies have been developed. Next-generation vaccines, based on mRNA and viral vector technologies, have recently emerged as alternatives to traditional vaccines. Adenoviral vector-based vaccines against coronavirus disease 2019 have demonstrated a more sustained antibody response as compared to mRNA vaccines. However, this has not been without complications, with a few cases of severe adverse events identified in vaccinated individuals, and the underlying mechanism is the subject of intense investigation. Adeno-associated viral vectors induce a weaker cellular immune response compared to adenoviral vectors, and it is mainly for this reason that there has been a diminished interest in exploring them as a vaccine platform until recently. This review will discuss recent developments and the potential of adeno-associated viral vectors as anti-viral vaccines.

## 1. Introduction

Pandemics caused by viral infections are a significant threat to human existence. It took about 90 years for a new vaccine to be developed after the first vaccine was developed in 1796 by Edward Jenner to prevent smallpox virus infection [1,2]. The scarcity in new vaccine development was because of slow progress in scientific advances. However, this has changed in recent years with discoveries and inventions in molecular biology, resulting in the rapid development of multiple vaccine design approaches. Vaccines based on inactivated or live-attenuated viruses and immunogenic proteins are currently the gold-standard. Recently, the nucleic-acid-based vaccines became an attractive alternative, with mRNA and adenoviral vector (AdV) vaccines being approved for use in several countries against coronavirus disease 2019 (COVID-19) [3]. However, there are limitations that come with these two vaccine technologies, such as short-lived antibody responses and/or adverse events. Most importantly, mRNA vaccines use non-viral delivery systems, e.g., lipid nanoparticles (LNPs). While these are generally safer and the packaging capacity is not as restrictive as compared to viral vectors, they result in poor transduction efficiencies, hence requiring the repeated administration of high doses [4,5]. This has driven the vaccine development field to explore alternatives.

Since their discovery in 1965, Adeno-associated viruses (AAVs) have been the most prominent gene delivery vehicles, with their strong efficacy and safety profiles being well-established in gene therapy applications. Low immunogenicity and long-term transgene expression are some of the key features that supported AAVs to succeed in several gene therapy clinical studies, with several therapies approved by the Food and Drug Administration (FDA) [6,7,8]. AAV is a 22 nm, non-enveloped virus that has a single-stranded, linear DNA genome of approximately 4.7 kb in length, and it depends on a helper virus for replication [9]. The AAV genome contains two inverted terminal repeats (ITRs) flanking three promoter elements (p5, p19, p40) and two genes, *rep* and *cap*, with the former responsible for viral replication and packaging, while the latter encodes capsid proteins and the assembly activating protein (AAP, Figure 1A). AAP directs the capsid proteins to the nucleus for capsid assembly. Recently, a sequence within the VP1 open reading frame has been mapped to the production of membrane-associated accessory protein (MAAP), whose function is yet to be defined [10]. Currently, 12 different AAV serotypes have been identified, with serotype 2 being the most investigated. AAV serotypes 2, 3, 5, and 6 are of human origin and have broader tissue tropism, whereas AAV serotypes 1, 4, and 7–12 are of non-human origin and have more specialized tissue tropism [11,12].

AAV-derived vectors (rAAVs) can be generated by replacing the *rep* and *cap* genes with a transgene of interest [13,14]. rAAVs mainly exist in two types: self-complementary AAVs (scAAVs) and single-stranded AAVs (ssAAVs). The transgene expression of scAAVs is quicker than ssAAVs, as the ssAAV genome requires the conversion to double-stranded DNA before transcription. The transgene capacity of scAAVs is 2.35 kb, which is half the size of ssAAVs (Figure 1A) [15,16]. The common workflow for rAAV production uses three plasmids, the AAV genome bearing plasmid engineered to carry the transgene, the AAV *cap* bearing plasmid and the helper plasmid bearing the adenoviral genes. These are co-transfected into HEK293T cells. Following viral replication, both the cells and supernatant are harvested for cell lysis and concentration by polyethylene glycol (PEG) precipitation, respectively. The clarified lysate and the precipitated virus are mixed and purified by iodixanol gradient ultracentrifugation and/or chromatography. Following quantification by q-PCR and purity analysis by polyacrylamide gel electrophoresis (PAGE), the vector can be used for in vitro studies or animal experiments, where intramuscular injection is the most commonly used route (Figure 1B) [17].

Similar to the wild-type AAV, rAAVs enters the cell by attaching to serotype-specific primary and secondary receptors on the cell surface [18,19]. AAVR (adeno-associated virus receptor), a type I transmembrane protein, is known to stabilize the attachment of most AAV serotypes to the cell surface [20]. This facilitates internalization, mainly by clathrin-mediated, caveolin-mediated, or clathrin-independent endocytic pathways. Once inside the cell, rAAV particles are sorted through early endosomes (EEs), late endosomes (LEs), and the Golgi apparatus [18]. Efficient transduction requires trafficking to the Golgi. Exposure to an acidic environment in the endosome allows for viral escape into the cytoplasm. If this fails, AAVs can be directed to lysosomes for degradation. Endosomal escape is followed by translocation to the cytosol before entering the nucleus via nuclear pore complexes, often facilitated by importin-β. Some rAAV vectors are marked with ubiquitin, signaling their degradation by the proteasome. Once in the nucleus, the AAV genome is released from the capsid. The single-stranded DNA genome is converted into double-stranded DNA, which can serve as a template for the synthesis of transcripts that are translated to rep proteins. Rep proteins mediate genome replication, and newly synthesized double-stranded genome copies serve as templates for the synthesis of RNAs that are translated into capsid proteins. This is followed by viral assembly and release (Figure 2) [18]. 

Development of AAVs as anti-viral vaccines was initiated following the first report of an AAV-based gene delivery vector in the 1980s; however, the interest in this diminished over time, until recently [21]. Hence, this review will discuss the application of AAVs as anti-viral vaccines during the last 10 years (Table 1). A recent study demonstrated that vaccination of mice with AAV6 designed to express tumor-associated antigens (TAAs) in dendritic cells (DCs) could increase cytotoxic T cells’ (CTLs’), CD8+ T cells’, and NK cells’ infiltration into the tumor microenvironment, leading to antigen-specific antibody production, the complement-dependent killing of melanoma cells, and protection of mice against tumor nodules in the lungs, highlighting the efficacy of the rAAV-based vaccines [22]. In a malaria vaccine study, AAV1 vectors were used as a boost immunization following priming with a human adenovirus type 5 (AdHu5) vector expressing the same antigens. The AdHu5-AAV1 regimen induced significant production of IFN-γ by CD8^+^ T cells following peptide stimulation and elicited high titers of antigen-specific IgG antibodies, which persisted at high levels for up to 280 days post-boost. Most importantly, up to 80% sterile protection against challenges with transgenic plasmodium sporozoites was observed [23]. However, the immunogenicity required to protect against new viral infections may differ. A potent anti-viral vaccine must induce both cellular and humoral immune responses to prevent progression to disease and provide a comprehensive long-lasting protection.

## 2. Immune Responses to AAV

Upon infection, AAVs induce the host’s innate and adaptive immunity in a mechanism similar to most viruses, as has been extensively reviewed before [40,41,42]. Briefly, AAVs bear pathogen-associated molecular patterns (PAMPs) that make them recognizable by pathogen recognition receptors (PRR) in a milieu of cells. These PRRs include Toll-like receptors (TLRs) that activate an innate immune response. In a cascade of signaling events, the interaction of PAMPs with PRRs induces the expression of pro-inflammatory cytokines or a type 1 interferon response, maturation of dendritic cells to antigen-presenting cells, and activation of other immune cells such as natural killer cells and macrophages [43,44]. Antigen presentation to CD8+ T and CD4+ T cells via the major histocompatibility complex (MHC) activates the adaptive immune responses. Activated CD4+ T cells differentiate into Th1 cells to mediate cytotoxic CD8+ T-cell responses and Th2 to mediate antibody responses. This eliminates antigens and establishes an immunological memory [43,45,46]. AAV capsid-specific T-cell responses are not commonly seen in animal models but are detected in human tissues. These T-cell responses have shown varied outcomes, from negligible effects to mild inflammatory effects manageable by immune suppressants or attenuation of transgene expression and subsequent reversal of the therapeutic effect [47,48,49,50,51,52]. Also, T-cell responses to AAV are less frequent in the young (<5 years of age) compared to adults. When they are observed, mainly in adults, T-cell responses appear as a memory phenotype, consistent with pre-exposure of individuals to the AAV capsid [53].

Several studies have reported the prevalence of pre-existing neutralizing antibodies (NAbs) against wild-type AAVs in the human population [54,55,56]. As with T cell responses, the relative serum concentration of pre-existing NAbs against AAV is relatively lower in younger people, as compared to adults [57,58,59]. The anti-AAV immunoglobulin G (IgG), more so IgG1, is represented and found to correlate with anti-AAV NAbs in healthy individuals [60,61]. These capsid-specific neutralizing antibodies display a high degree of longevity and cross-reactivity owing to the presence of highly conserved regions among different wild-type AAV serotypes, including some synthetic AAV vectors [62,63,64]. The low immunogenicity of AAVs as compared to other viral vectors has been dubbed the downfall in vaccine development. Hence, efforts have been made to improve AAVs’ immunogenicity by modifying both the genome and the AAV capsid.

## 3. AAV Modification to Improve Immunogenicity

### 3.1. Genome Modification

The delayed transgene expression kinetics from the single-stranded genome of rAAVs are believed to allow epitopes in the capsid to have an adjuvant effect, which is then boosted by encoded antigen-specific immunity a few weeks later. The double-stranded property of scAAV genomes is an advantage because it makes the genome recognizable by TLR-9 to induce a stronger adaptive immune response [65]. It is well established that the presence of unmethylated CpG islands in AAV genomes enhances immunogenicity [66]. Factors that influence the antigenicity of transgene products encoded by AAVs include the inherent immunogenic properties of a transgene, pre-existing immunity to the encoded antigen, antigen presentation and processing, route of delivery, AAV serotype, dose, and efficient transduction of dendritic cells [67,68,69,70].

Expression cassettes have been engineered to include signal sequences, such as the MHC class I trafficking signal and the IgE secretary signal, and immunogenic proteins such as ovalbumin. Adding these signal peptides resulted in improved antigen folding, endoplasmic reticulum (ER) targeting, processing, and presentation on MHC molecules, inducing a robust cellular immune response against the antigen [71,72]. To improve the immune response against the transgene product, Wu and their team designed the AAV6 vector to express a modified receptor-binding domain (RBD) of the SARS-CoV-2 spike protein fused to a tissue plasminogen activator (tPA) signal peptide, a T4 fibritin-derived trimeric motif to mimic the natural trimeric viral conformation of the RBD and RS09, a TLR-4 agonist to serve as an adjuvant. An improved activation of CD8+ T cells against the transgene product was observed [24,68]. An AAV displaying a cell-penetrating peptide (AAV-ie) was designed to express various truncated spike (S) proteins of SARS-CoV-2, which were fused to an IL-2 lead signal to enhance antigen secretion and the T4 fibritin-derived trimeric motif. Intramuscular injection of 1 × 10^13^ vg of this vector induced high S-binding antibody levels in macaques, which were maintained over two months post-vaccination. The immune sera from non-human primates (NHPs) showed pseudo-virus-neutralizing activity against SARS-CoV-2 variants B.1.351 and B.1.1.7. T-cell responses were evaluated by stimulating peripheral blood mononuclear cells (PBMCs) before and 8 weeks post-vaccination with an S peptide pool. Increased populations of IFNγ+, IL2+, TNFα+, CD4+, and CD8+ T cells were observed in PBMCs post-vaccination (Figure 3) [25,73].

### 3.2. Capsid Modification

Insertion of major histocompatibility complex class I (MHC I)- or MHC II-restricted epitopes in a capsid of AAV2 expressing immunogenic ovalbumin has been demonstrated to result in superior transgene expression, transgene-specific immune responses, and protection of mice against tumor development in a tumor cell challenge model as compared to AAV2 with an unmodified capsid [72]. AAVs’ poor transduction efficiency of dendritic cells (DCs) is speculated to contribute to AAV’s feature of being relatively non-immunogenic. This realization contributed to the modification of the AAV vector capsid to optimize DC transduction. Several studies demonstrated that inserting peptides and substituting single amino acids into the AAV capsid improved DC transduction. Insertion of short peptides such as VSSTSPR, ISSSTAR, and NNPLPQR at position 587 of the capsid ORF conferred improved DC transduction and increased the uncoating capacity of the vectors. Improved intracellular processing with higher levels of episomal DNA in immature and mature DCs post-transduction was also observed. Intramuscular injection of these peptide-decorated vectors in mice resulted in higher levels of antigen-specific IgG antibodies and robust CD8+ T-cell responses [74,75]. Serine substitution at position 663 in the VP3 capsid protein with valine improved stability, ability to interact with target cells, intracellular trafficking, nuclear translocation, and DC targeting. This modification resulted in a robust immune response characterized by relatively high levels of antigen-specific CD8^+^ T cells and IFN-γ-producing cells (Figure 3) [24,68,71].

## 4. Recent Studies Demonstrating the Potential of rAAVs as Vaccines Against Viral Infections

### 4.1. Hepatitis E Virus

A chimeric myotropic synthetic AAV capsid (AAVMYO3) was designed to carry a yellow fluorescent protein encoding gene and open reading frame 3 (ORF3), encoding for a small multifunctional phosphoprotein essential for hepatitis E virus (HEV) infection in vivo [76]. Specifically, the ORF3 gene was cloned into an scAAV vector, and an untranslated 450 bp fragment of the YFP sequence was inserted after the BGH polyA terminator to adjust the length of the entire insert between the inverted terminal repeats (ITRs) to the optimal ~2000 bp. AAVMYO3 was administered intravenously to young (6-week-old) female BALB/c mice at 1 × 10^11^ or 1 × 10^12^ vg per mouse, and the vector’s biodistribution was localized in the muscle where antigen expression was detected and not in the spleen or liver. The liver is known for inducing tolerance to delivered gene therapy, and a lack of vaccine expression in this tissue might be desirable [31]. Anti-ORF3 antibodies were detected in vaccinated mice, with antibody production correlating with ORF3 expression and peaking at six weeks post-injection. The induced antibodies showed a moderate neutralization effect on HEV particles in vitro. The study highlighted the potential of AAVMYO3 for inducing ORF3 expression and anti-ORF3 antibody production, suggesting further research directions such as booster strategies and in vivo challenges.

### 4.2. Hepatitis C Virus

Rhesus-macaque-derived hybrid vectors AAV2/rh32.33 and AAV2/8 were utilized to express the envelope glycoprotein E2 from the hepatitis C virus (HCV) genotype 1b as candidate vaccines against HCV infection. Vectors were administered intramuscularly to young (4–6 weeks old) C57BL/6 mice at a dose of 1 × 10^11^ vg per mouse. The AAV vaccines induced a strong and persistent increase in E2-specific antibody production over time. Antibody levels peaked at 16 weeks post immunization with the AAV2/rh32.33 vector showing particularly favorable immunogenicity. Both AAV vaccines demonstrated significant neutralization of HCV pseudo-particles (HCVpp) derived from genotypes 1a and 1b at week 12 post-immunization. The cross-neutralization ability against genotypes 1a and 1b was superior to that against genotypes 2a, 3a, and 5a. Serum samples from humans showed lower neutralizing antibody titers against AAV2/rh32.33 compared to AAV2/8, highlighting the potential of evading pre-existing immunity to naturally occurring AAV serotypes [32].

### 4.3. Influenza Virus

A recent study developed several AAV9 vectors expressing the hemagglutinin (HA) or nucleoprotein (NP) of the Cal/7/9 (H1N1)pdm virus [33]. These included AAV-HA vector expressing the wild-type HA protein and AAV-NP expressing the wild-type NP protein. An intranasal dose of AAV9 at 10^11^ vg and 7.5 × 10^12^ vg were administered to mice and ferrets, respectively. Among the tested vaccines, the AAV-HA vector performed the best overall. AAV-HA induced strong antibody responses against H1N1 viruses, including the 1918 pandemic H1N1 virus and H5N1. It also induced HA-head- and HA-stalk-specific antibodies, providing a broader range of protection. AAV-HA offered complete protection against a homologous Cal/7/9 challenge. It also significantly reduced weight loss and protected against the heterologous A/Puerto Rico/8/1934 H1N1 challenge. In ferrets, AAV-HA showed reduced disease severity, lower virus replication, and less lung pathology than other groups. It induced high levels of neutralizing antibodies, which correlated with reduced viral titers in the respiratory tract. AAV-HA induced broadly reactive FcγR-activating antibodies, crucial for anti-viral effector mechanisms.

### 4.4. HIV

The successes and limitations observed in the administration of recombinant anti-HIV monoclonal antibodies (mAbs) for immunoprophylactic and immunotherapeutic purposes have prompted further studies on this topic. AAVs are an attractive option to simplify vaccine production and prolong immunological responses [77,78]. AAV8 expressing full-length NAbs targeting the HIV-1 envelope protein was previously designed. Sustained antibody expression was observed following intramuscular administration in immunocompromised mice, with secreted antibodies maintaining their neutralizing activity [37]. To increase tissue penetration and reduce off-target effects that may come with full-length antibodies, van Dorsten et. al. engineered AAVs to express single-chain fragments bearing only the antibody variable region of the light and heavy chains targeting the HIV-1 envelope. Although other versions of the single-chain variable fragments (scFvs) lost neutralizing activity depending on the key residues within the epitopes, the authors were able to design scFVs that maintained significant potency and substantial breadth against diverse global strains of HIV [38]. Meanwhile, Gardner et. al. developed an AAV vector expressing an antibody-like SIV or HIV-1 entry inhibitor called eCD4-Ig. eCD4-Ig simulates the HIV receptor and blocks its interaction with host cell receptors and infection. Intramuscular injection of AAVs in rhesus macaques resulted in full protection against the SIV challenge [39].

### 4.5. Ebola Virus

The potential of passive vaccination was also highlighted by Ebola virus studies. AAV9 vectors expressing one of the three mAbs against the Ebola virus surface glycoprotein were developed. Mice were vaccinated by intramuscular or intranasal administration of 10^11^ vg of AAVs before an intranasal challenge with mouse-adapted Ebola virus two weeks later. About 83% protection was observed in mice injected with one AAV or multiple AAVs expressing different antibodies intranasally [34]. Intramuscular delivery of AAV2 encoding non-neutralizing or neutralizing antibodies against the Ebola virus was followed by a challenge with mouse-adapted Ebola virus 1 week or 5 months following vaccination. Interestingly, 100% and 83% protection were observed in mice that received AAV2 expressing non-neutralizing and neutralizing antibodies, respectively. Co-administration of these vectors further improved their protection efficacy [35]. A recent study designed three AAV6.FF vectors, an AAV6 designed to carry lung and muscle transduction-enhancing capsid mutations, expressing Ebola-virus-specific mouse- or human-derived mAbs. Administration of these vectors intramuscularly or intranasally resulted in the expression of antibodies for the mouse’s lifetime and protected mice from a mouse-adapted Ebola virus [36].

### 4.6. SARS-CoV-2

As with other gene delivery platforms, rAAVs were extensively explored for their potential use as vaccines against SARS-CoV-2, and only selected studies are discussed here. Five AAV5-based vectors were designed to encode the receptor-binding domain (RBD) and other S protein regions: ssAAV5-RBD, ssAAV5-RBD-plus (RBD + 42 residues), ssAAV5-S1, ssAAV5-S, and ssAAV5-NTD. The vectors also expressed IgE secretory signal peptides to direct antigens to the endoplasmic reticulum. The single-dose vaccination with 1 × 10^11^ vg of rAAV5-based vaccine induced robust immune responses in mice, including high levels of neutralizing antibodies and functional T-cell responses. The vaccines induced high levels of neutralizing antibody titers against circulating SARS-CoV-2 variants, including Alpha, Beta, Gamma, and Delta, and were at a peak level of over 1:1024 for more than a year post-injection. The rAAV5-based vaccines protected both young and old mice from SARS-CoV-2 infection in the upper and lower respiratory tracts. The scAAV-based vaccines elicited higher titers of IgG and neutralizing antibodies as compared to ssAAVs [30].

AAV9 is the most efficient gene delivery vector among the tested serotypes after local injection into the skeletal muscle of C57BL/6 mice [79]. Hence, the AAV9 vaccine expressing the RBD of the SARS-CoV-2 spike protein was developed and its immunogenicity evaluated in mice [26]. Each mouse was immunized with 1 × 10^11^ vg of the rAAV (AAV9-RBD). Eight weeks post-administration, all mice survived and maintained their weight. The AAV9-RBD vaccine induced the production of RBD-specific IgG antibodies with geometric mean titers (GMTs) of 1:4873 (intramuscular injection) and 1:5385 (nasal drip). Neutralizing antibodies against the SARS-CoV-2 pseudovirus were detected in all mice 8 weeks after immunization. The mean neutralizing antibody EC_50_ values were 517.7 ± 292.1 in the intramuscular injection group and 682.8 ± 454.0 in the nasal drip group. Levels of IFN-γ, IL-2, IL-4, and IL-10 were significantly increased in both the nasal drip and intramuscular injection groups compared to the negative control.

The AAV2/9 vector pseudotype was designed to express the stable RBD domain (SRBD) of SARS-CoV-2. Domain analysis of the spike protein identified the most stable location, and based on the findings, a β -sheet spanning from Q321 to S591 forming C and N tails of RBD was incorporated to enhance thermostability. The rhesus macaques received a single-dose immunization of 1 × 10^12^ vg/macaque (high), 1 × 10^11^ vg/macaque (middle), or 1 × 10^10^ vg/macaque (low) of AAV-SRBD. Parameters such as seroconversion rate, immune response, long-lasting immunity, cross-neutralization, T-cell response, safety, liver function, and efficacy against variants were evaluated. The seroconversion rate (antibody titer > 800) reached 100% at 35 days post-vaccination in macaques receiving the high and middle doses, but only 33.3% in the low-dose macaques achieved the same seroconversion. The AAV-SRBD vaccine demonstrated good immunogenicity in macaques receiving the high and middle doses. The SRBD antibodies remained high until 598 days post-vaccination in the high-dose macaques, indicating a long-lasting immune response. Neutralizing antibodies showed efficacy against the various SARS-CoV-2 variants being assessed, which included the Beta Variant (B.1.351), Delta Variant (B.1.617.2), and Omicron Variant (B.1.1.529), suggesting cross-neutralization capability and potential protection against emerging variants. The vaccine activated the RBD-specific T-cell responses. The AAV-SRBD vaccine was safe in the macaques, with no significant abnormalities in clinical physiology or evidence of experimental toxicity. Although AAV gene expression was detected in the liver, there was no evidence of liver dysfunction or inflammation post-vaccination; the vaccine did not trigger severe inflammation, and pathological indicators in the blood remained normal after immunization [27].

Two AAV vector vaccine candidates named AC1 and AC3 based on AAVrh32.33 were designed as vaccines against SARS-CoV-2. The AC1 vaccine candidate expressed a prefusion-stabilized, full-length S protein under the control of an SV40 promoter, with a short SV40 polyadenylation signal (poly-A). AC3 was designed to express the secreted S1 subunit under the control of a CMV promoter with regulatory elements consisting of a woodchuck hepatitis virus posttranscriptional regulatory element (WPRE) and the bovine growth hormone poly-A. In a challenge study, macaques were injected with 10^12^ vg/macaque of AC1 or AC3 before being challenged 8 to 9.5 weeks post-vaccination with 1 × 10^5^ plaque-forming units (pfus) of the SARS-CoV-2 virus, which was administered via a combination of intranasal and intra-tracheal routes. Rhesus macaques exhibited robust cellular responses to the AC1 and AC3 vaccines, with detectable T-cell responses persisting for up to 44 weeks post-vaccination. Rhesus macaques developed high levels of SARS-CoV-2 RBD-binding and neutralizing antibodies following vaccination with AC1 and AC3. Antibody levels remained elevated for at least 11 months post-vaccination, with neutralizing titers comparable to those seen in convalescent patients. The neutralizing antibody responses to AAVrh32.33 developed slowly and to relatively low levels. No cross-reactive neutralizing antibody responses were detected against AAV1, AAV2, AAV5, AAV8, or AAV9. The PET and CT scans for cynomolgus macaques revealed that vaccinated macaques had no significant lung lesions, but control animals showed evidence of lung pathology. PET scans showed increased activation of lung-draining lymph nodes in control animals post-challenge, indicating SARS-CoV-2 infection. Animals that received the AC1 vaccine did not display significant changes in inflammatory responses following the challenge, which is likely the result of effective neutralization of the virus. The scans also indicated reduced lung inflammation and viral replication in the vaccinated group, suggesting a protective effect of the vaccine against SARS-CoV-2 infection in the macaques [29].

AAVrh32.33 is a man-made vector; hence, it might have an intrinsic structural weakness that might reduce yields during production. To optimize first-generation AC1 and AC3 vaccine candidates for large-scale vaccine manufacture, seroconversion, and potency at low doses, Zabaleta and colleagues designed a second-generation vaccine platform (ACM1, ACM-Beta, and ACM-Delta) based on a naturally occurring AAV11 because of its structural similarity to AAVrh32.33 [28]. The second-generation vectors expressed codon-optimized, prefusion-stabilized, full-length spike proteins from different SARS-CoV-2 variants (including Wuhan, Beta, and Delta variants). To improve vaccine efficacy against various SARS-CoV-2 variants and potency at lower doses while maintaining robust immune responses, different promoters such as the short EF1a promoter (EFS), minimal CMV promoter (miniCMV), and full CMV promoter as well as synthetic polyA were used. These second-generation vaccines demonstrated superior immunogenicity, enhanced protection against specific variants, broader cross-reactivity, and improved antigen expression compared to the first-generation platform [80].

## 5. Discussion

The development of AAV vectors as anti-viral vaccines has made significant strides in recent years. Efforts to improve immunogenicity have made AAVs the next attractive vector for vaccine development. The single-stranded and double-stranded property of AAV genomes allows immune priming and boosting in one dose or activates the TLR-mediated immune response and augments the AAV vaccine development toolbox. Genome modifications that include the fusion of transgenes with sequences encoding peptides that can enhance TLR activity and improve antigen folding, secretion, processing, and presentation, decoration of the AAV capsid with peptides, and substitution of capsid amino acids to improve vector interaction with immune cells have proved useful. Both the genome and capsid modification have led to improved AAVs that can induce robust antigen-specific, T-cell-mediated and humoral responses and confer protection against viral infections [71,72].

As a result of the unique microenvironment and specialized immune cell types present in different tissues, the route of administration has significant implications for the vaccination outcome. A recent SARS-CoV-2 study used nasal drip and intramuscular rAAV vaccine administration in mice. Interestingly, intramuscular injection resulted in lower antibody levels, but with a higher neutralizing potency as compared to a nasal drip, whereas similar cytokine responses were observed [26]. In an Ebola virus vaccine study, intramuscular injection of immunocompetent mice with AAVs expressing mAbs resulted in no protection because of humoral immune responses against the mAbs, as compared to a complete protection observed when intranasal administration was performed [34]. The wealth of rAAVs derived from diverse AAV serotypes, modified capsids, and artificial capsids with unique tropisms will enable flexibility in the route of AAV vaccine administration [81]. Hence, the choice of the administration route can be guided by the unique immune signatures required for specific viral infection.

AAVs generally require a high dose to achieve a therapeutic effect, and this has previously resulted in deaths in non-human primates [82]. However, vector engineering has allowed the use of relatively lower doses to achieve vaccine efficacy compared to doses conventionally required for gene therapy [28,72]. A single vector administration simplifies the vaccination process by reducing the number of injections needed to achieve immunity.

During chronic pathogenic infections such as with hepatitis B virus (HBV), the constant viral load exposure of the host leads to aberrant inflammation, a tolerogenic immune environment, and immune dysfunction [83,84]. Hence, persisting antigen expression from AAVs, which leads to constant activation of immune signaling pathways, raises concern. Data showing that sustained antigen expression is critical for the durability of protection observed in rAAV vaccine studies are scarce. However, the use of AAV-based vaccines with enhanced immunogenicity and at doses lower than those used in gene therapy studies may result in clearance of transduced cells earlier than when unmodified rAAVs are used [28,72,74]. Whereas this long-term transgene expression may be a challenge when expressing foreign proteins, it is attractive for passive immunization. Long-term expression of human-derived mAbs from rAAVs may enable sustained immunity, which is of particular interest during a pandemic, as it may guard against infection with the emerging viral variants. In addition, studies showing transgene expression accompanied by persistent antibody responses in AAV vaccinated animals without any adverse events are promising [27,28,29]. However, safety studies investigating the effects of possible long-term AAV-mediated antigen expression on the immune system in the context of vaccine administration to animals and humans are necessary.

## 6. Conclusions

The future remains unknown, but new and deadly viral strains as aggressive as SARS-CoV-2 or even worse might emerge. AAV’s ability to safely deliver antigens and persist in cells, requiring only a single shot to confer comprehensive protection against a pathogen, including several emerging variants, is a valuable property that motivates the continued development of AAVs as a vaccine platform.

## Figures and Tables

**Figure 1 pharmaceutics-16-01360-f001:**
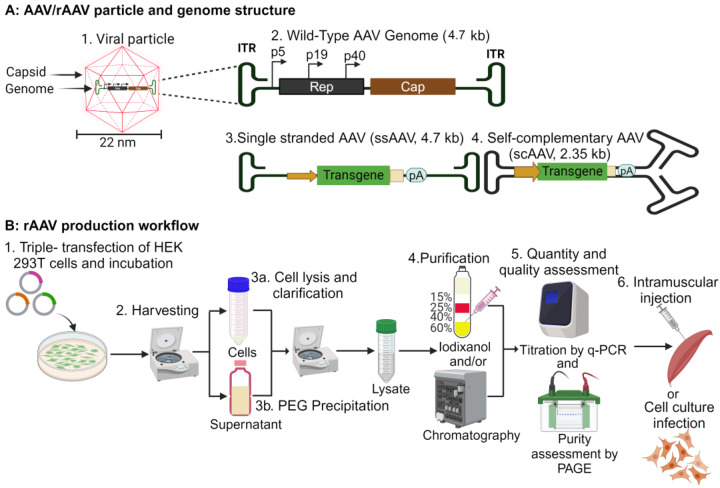
AAV/rAAV structure and production workflow. (**A**) The particle structure is made of a 22 nm capsid that encapsulates a 4.7 kb single-stranded DNA genome (1). The genome bears two inverted terminal repeats (ITRs) that flank the *rep* and the *cap* open reading frames. Gene expression is driven by three promoters (p5, p19, and p40, 2). The rAAV genome can be in an ssDNA form (ssAAV) with a packaging capacity of 4.7 kb (3) or a self-complementary DNA form (scAAV) with a packaging capacity of 2.5 kb (4). (**B**) The rAAV workflow generally involves a triple transfection of HEK293T cells and incubation to allow viral replication (1). The cultures are centrifuged to harvest the cells and the supernatant (2). The cell pellet is lysed and clarified by centrifugation (3a), and polyethylene glycol (PEG) precipitation of the virus from the supernatant is performed to concentrate the virus (3b). The clarified lysate is used to resuspend the PEG pellet. The lysate is then purified by iodixanol gradient ultracentrifugation and/or chromatography (4) before titration by q-PCR and purity assessment by polyacrylamide gel electrophoresis (PAGE, 5). The purified rAAV is used for in vitro and in vivo studies where intramuscular injection is commonly used for vaccine studies (6) (created in BioRender.com).

**Figure 2 pharmaceutics-16-01360-f002:**
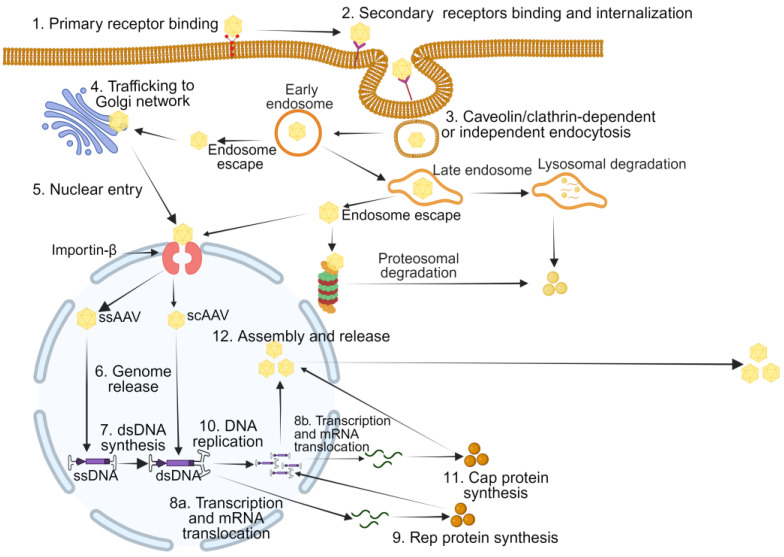
rAAV replication cycle. The rAAV capsid binds to primary receptors on the cell surface (1), facilitating interaction with secondary co-receptors (2) for internalization via clathrin-mediated, caveolin-mediated, or clathrin-independent endocytosis (3). Following endocytosis, rAAV particles escape the early (EEs) or late endosomes (LEs) to the cytosol before targeting the Golgi apparatus (4) or nucleus directly (5). In the event of failure to escape, rAAVs can be degraded by the lysosomes or proteasomes. Once in the nucleus, the rAAV genome is released from the capsid (6). For ssAAVs, the single-stranded DNA genome is converted into double-stranded DNA (7), while the scAAV genome can directly serve as a template for transcription (8a) followed by rep protein synthesis (9). Rep proteins then mediate genome replication (10) with the newly formed double-stranded genome copies serving as the major template for synthesis of cap proteins encoding RNA (8b), followed by rep protein synthesis (11), viral assembly, and release (12) (Created in BioRender.com).

**Figure 3 pharmaceutics-16-01360-f003:**
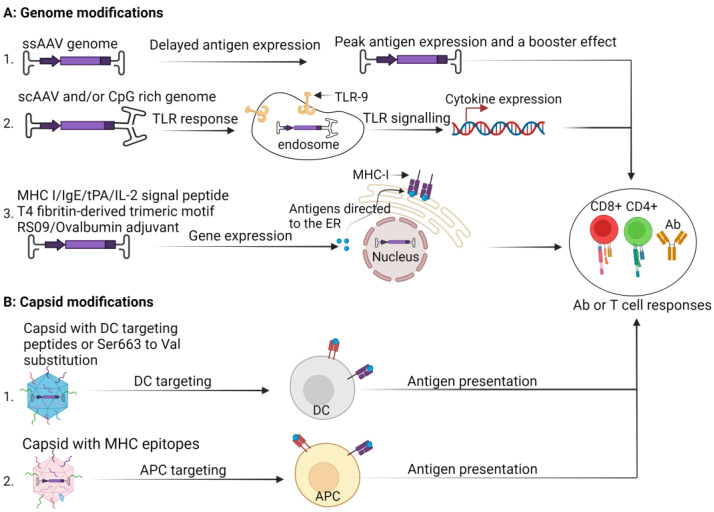
AAV modifications to enhance immunogenicity. (**A**) Genome modifications. AAVs carrying single-stranded AAV (ssAAV) allow the capsid to act as an adjuvant before ssAAV achieves full antigen expression to boost the vaccine effect (1). Self-complementary AAVs or any genome conformation carrying unmethylated CpG islands are recognizable by Toll-like receptor-9 (TLR-9) receptors, whose signaling leads to pro-inflammatory cytokine response (2). Incorporation of sequences encoding various signal peptides, including MHC-I trafficking signal peptide, IgE/IL-2 secretory domain, and tissue plasminogen activator (tPA), or the T4 fibritin-derived trimeric motif or adjuvants, such as RS09 and ovalbumin, direct antigens to the endoplasmic reticulum (ER), where peptides bind to MHC I for antigen presentation or to improve antigen folding/secretion and/or boost overall immune responses (3). (**B**) Capsid modification. Capsid-bearing dendritic cell (DC) targeting peptide or S663V point mutation transduces DCs efficiently (1). Capsids with MHC epitopes target antigen-presenting cells (APCs) (2). Overall, all the modifications enhance antibody (Ab) and T-cell responses (created in BioRender.com).

**Table 1 pharmaceutics-16-01360-t001:** Preclinical AAV vaccine studies evaluating efficacy.

AAV Serotype	Target	Antigen/Antibody	Reference
AAV6	SARS-CoV-2	RBD	[24]
AAV-ie	SARS-CoV-2	Stabilized RBD (SRBD)	[25]
AAV9	SARS-CoV-2	RBD	[26]
AAV2/9	SARS-CoV-2	SRBD	[27]
AAVrh32.33 and AAV11	SARS-CoV-2	Spike	[28,29]
AAV5	SARS-CoV-2	S1/RBD	[30]
AAVMYO3	HEV	ORF3	[31]
AAV2/8	HCV	E2	[32]
AAV9	Influenza	HA and NP	[33]
AAV9	Ebola virus	Full-length mAbs	[34]
AAV2.FF	Ebola virus	Full-length mAbs	[35].
AAV6	Ebola virus	Full-length mAbs	[36]
AAV8	HIV	Full-length mAbs	[37]
AAV8	HIV	Single-chain variable fragment	[38]
AAV1	HIV	Antibody-like entry inhibitor	[39]

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
