# Peer review of "Recent Advances in Designing Adeno-Associated Virus-Based Vaccines Against Viral Infections"

_pharmaceutics, 2024, doi:10.3390/pharmaceutics16111360_

Round 1
Reviewer 1 Report
Comments and Suggestions for Authors
Njabulo Mnyandu et al. discuss recent advancements in adeno-associated virus (AAV) vectors for anti-viral vaccines, highlighting how enhancements in immunogenicity make AAVs an attractive option for vaccine development. Their single-stranded and double-stranded genome properties facilitate effective immune priming and response activation. This article is compelling and potentially significant; however, a few important points need to be addressed.
1) The persistent expression of AAV antigens raises concerns about chronic immune activation, especially in the context of ongoing infections such as hepatitis B. The article should provide a more contextual explanation of this issue to engage the research community better and underscore its significance in relation to vaccination strategies.
2) [ Lines 92-97] Please cite references.
3) The discussion section should be enhanced to specifically address advancements in AAV vaccination strategies against pathogens and their correlation with the onset of immune responses. Emphasizing this aspect will clarify how these vaccination strategies can contribute to disease control and improve overall vaccination approaches.
Reviewer 2 Report
Comments and Suggestions for Authors
Line 33: "life-attenuated viruses" should be "live-attenuated viruses."
The word "anti-viral" is inconsistently hyphenated. Use either "antiviral" or "anti-viral" consistently.
Line 15: "Adenoviral vector-based vaccines against coronavirus disease in 2019..." – It would be clearer to specify "coronavirus disease (COVID-19)" instead of "coronavirus disease in 2019."
Line 49-50: When listing AAV serotypes, the distinction between human and non-human origins could be better explained.
AAV vectors tend to persist in cells, leading to long-term expression of the transgene. While this is advantageous for generating long-lasting immunity, it may also result in constant immune activation against the antigen, which can lead to chronic inflammation or immune-mediated damage. Can authors summarize the best strategy to overcome this issue?
Reviewer 3 Report
Comments and Suggestions for Authors
This review article (“Recent advances in designing AAV-based vaccines against viral infections”) by Mnyandu et al. discusses about the recent developments and the potential of Adeno-associated viral vectors as antiviral vaccines. The article seems to be interesting, and is quite informative as well. However, I have a few suggestions to the authors before I can recommend it for the publication in the journal Pharmaceutics and I think incorporating them would definitely improve the article. My suggestions are listed below:
1. I believe that this review article can significantly benefit from having a few schematic figures. Having such figures would definitely help to attract a broader audience. These would assist the readers easily understand complex information, such as complex processes, or phenomena that are discussed in the text and also make the article more attractive. In that direction, I have a few suggestions:
(a) Comparisons between non-vectored and vectored immunotherapeutic strategies.
(b) An introductory schematic representation of the AAV genome and vector (including particle structure) – with proper labelling of each of the segments/components.
(c) Diagram of rAAV transduction pathway including the cell attachment and entry.
(d) Construction/Production of Adeno-associated Vectors. And finally, delivery routes.
With all these background information, I believe the current Figure 1 would be more meaningful.
2. The present form of the article is limited to only a few viral infections (HEV, HCV, and SARS-CoV-2). There are also recent examples for other viruses, such as HIV, Flu, Ebola, Influenza, RSV etc. I would suggest the authors to include discussions about those (with related references), given the broad title of the article.
A few minor issues:
1. Page 2, Line 52: AAV serotype 12?
2. Page 5, Line 158: “…poor transduction efficiency of DCs is…” The full-form of DC should be included at its first appearance in the article.
3. Page 7, Line 246 and Line 247: Should there not be a unit associated with the mean neutralising antibody EC50 values?
4. Page 7, Line 250: “…pseudotype was designed to express the stable RGB domain…” RBD domain?
Round 2
Reviewer 3 Report
Comments and Suggestions for Authors
The authors (Mnyandu et al.) have made a sincere effort to answer most of my queries, as well as those of the other reviewers. They have done quite a bit of additional writing with appropriate literature support in response to the raised concerns. As a result, the quality of the article after the revision, has improved significantly. I, therefore, strongly recommend the acceptance of the review article for publication in the Pharmaceutics journal.
A few minor issues:
1) Please use only one style of the short form for SARS-CoV-2.
2) Page 3, Line 189: Polyethylene glycol (PEG)
3) Page 5, Line 303: Nucleus in place of nuclear?
4) Once the short form has been introduced, please use IFN-γ afterwards.
5) Page 12, Line 812-813: If the authors are unsure about the unit (for EC50 values), please use the same style that has been used in the cited paper.
6) Page 12, Line 818: ‘β-sheet’ in place of ‘b-sheet’?
Once the authors have made these small corrections, the review article can be accepted for publication.
